# The Use of Daily Growth to Analyze Individual Spawning Dynamics in an Asynchronous Population: The Case of the European Hake from the Southern Stock [†]

Cristina García-Fernández [1,*], Rosario Domínguez-Petit [2] and Fran Saborido-Rey [1]

1    Instituto de Investigaciones Marinas (IIM-CSIC), 36208 Vigo, Spain
2    Centro Oceanográfico de Vigo (IEO-CSIC), 36390 Vigo, Spain
*    Correspondence: cgarcia@iim.csic.es
†    The paper was rewritten/reproduced from author's works in PhD's thesis and has not been published.

**Abstract:** Daily growth patterns and their relationship with reproduction was analyzed in the European hake from the Galician Shelf, where it shows a very protracted spawning with three spawning peaks. The daily growth analysis was performed in otoliths of adult females on the transversal section of the sagittae otolith. Daily increments were measured from the border to the nucleus in females until they were discernible. Results show that daily growth of females decreases during the spawning period because they allocate less energy to somatic growth in favor of the production of gametes, with an increase in growth in July. Lastly, daily growth individual trends showed a "spawning pattern" in 28% of medium and large females, suggesting an individual spawning period of one to two months, with 4–5 valleys of narrow daily increments, likely associated to batch release: individual spawning frequency would be 4–5 days. This is the first time that individual spawning frequency in hake is estimated based on individual data. Finally, the spawning pattern is detected only once per year, indicating that a single female participates only in one spawning peak per year, supporting the hypothesis of the existence of two or more spawning components in the stock.

**Keywords:** trade-off; daily growth; spawning; otolith; European hake





## 1. Introduction

Energy devoted to growth and reproduction is balanced along the whole lifetime of an individual hake [1]. Energetic resources are used for maintenance and somatic growth when the fish is immature. However, when it reaches sexual maturity, the energy allocation changes: the surplus energy, which was previously only used for growing, is redistributed between growth and reproduction [2,3]. Part of this energy is transferred for reproduction processes (gamete production and reproductive behaviour) with the consequent growth rate decrease. The seasonality on growth (intra-annual variability) in fish is also triggered by physiological events like spawning, but also by environmental conditions [4].

This trade-off is strongly related to the energy acquisition strategy of the species. In capital breeders, the growth rate is expected to be low during spawning season because of feeding reduction during this period. If this is so, a reduction in growth rate should be traceable in the otolith daily growth pattern [5]. On the other hand, in income breeder species, there is not an interruption of feeding while spawning that could mask the influence of reproduction on growth [6–8].

Based on this trade-off, it could be interesting to study the daily growth to validate whether the individual reproductive activity can be monitored on daily otolith growth patterns. In fact, it would be very useful in exploited marine species with protracted spawning seasons and high population asynchrony in spawning due to the associated

temporal dispersion on the estimation of reproductive parameters (spawning fraction, spawning frequency, etc.).

Thus, analyzing daily growth in the otolith could provide individual information on species where behavior at the population level hinders the estimation of parameters. The target of the study is the European hake, *Merluccius merluccius*. This species can be found in temperate areas of the Eastern Atlantic, principally throughout the Northeast Atlantic Ocean and the Mediterranean Sea, from Norway to Iceland in the north and to Mauritania in the south [9,10]. Along its distribution, the species presents geographical variability in different aspects, such as reproductive traits [11], i.e., size at maturity [12], diet [13,14], or spawning season [12]. This study is focused on the population located in the Galician shelf (NW Iberian Peninsula), the main spawning area in the southern stock [15–17]. It is a very plastic species that adapts its reproductive potential to environmental conditions, energy availability, and fishing pressure [18,19]. This capacity to adapt well is even more evident in highly productive environments such as the Galician shelf, where important upwelling events take place [20]. Due to this, in the study area, the species presents a protracted spawning season with high-population reproductive asynchrony and an income breeding strategy. Despite the protracted spawning season, three spawning peaks are clearly identified: the main one in winter−spring, the secondary one in summer and occasionally another one in autumn. However, there is great interannual variability in both intensity and timing [10,21,22], likely because of the influence of several internal and external factors such as environment, food availability, or even growth.

Based on its population asynchrony, key individual parameters such as spawning duration, spawning fraction, or spawning frequency are not known for this species. Therefore, the main objective of this study is to test the use of daily growth patterns to extract reproductive parameters at the individual level in one species with high population asynchrony. In relation to this objective, the daily growth of the adult females was analyzed to elucidate if there are changes in the female physiology due to the reproductive cycle, and specifically spawning. If the trade-off between spawning and growth is detectable, it would be possible to estimate reproductive parameters at individual scale in the spawning stock of *M. merluccius* in the Galician Shelf. To do so, the daily growth of mature females was studied. Thus, the objectives of the study are to test if (i) the daily rings formed in the otoliths during the spawning peaks are narrower than those rings formed in the rest of the year, (ii) there are differences in the magnitude of this narrowing between spawning peaks (i.e., there are differences in energy investment in each spawning peak), and (iii) it is possible to estimate individual reproductive parameters based on this method.

## 2. Materials and Methods

The approach of the present study was created on the analysis of daily growth in otoliths of adult females. Based on the assumption that actively spawning females feed less and devote more energy to reproduction than growth, a decrease in daily growth during the spawning peak is expected, i.e., narrow daily increments.

Based on the reproductive season, and assuming that during the spawning period, the daily growth of an active female decreases, each female would present a low daily growth pattern once perceived in each female's otoliths if it participates in one spawning peak. On the contrary, if each female spawns in both of the two main spawning periods, that pattern should appear two or more times in each female's otoliths. Finally, if spawning activity does not impact daily growth, no detectable pattern in the otoliths would be expected (Figure 1).

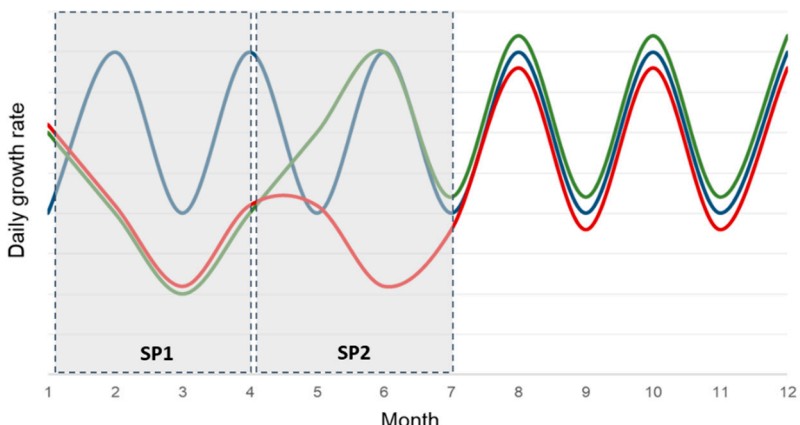

**Figure 1.** Plot for describing the three potential patterns of the daily growth of adult females of European hake, assuming lower growth when active spawning takes place (i.e., narrow daily rings in otoliths). Grey boxes: each spawning peak (SP1: winter-spring and SP2: summer). Red line: assuming that each female spawns in the two main spawning periods; Green line: assuming that each female participates only in one spawning peak (in this plot a female actively spawning only in SP1); and blue line: assuming that daily growth is not related to spawning activity.

To carry out this study, 60 adult females were selected from the monthly sampling of the commercial gillnet fleets on the Galician shelf during the whole of 2017. This sampling effort was higher during the first half of the year to cover the two main spawning peaks. Data recorded from sampled females included the total length (TL) (cm), the total, gutted, gut, liver, and gonad weights (g), as well as sex and macroscopic ovary stages. Selected females were sampled between the two main spawning peaks (April–May), and after the second spawning peak (July), and the total length ranged from 34 to 80 cm.

The daily growth increments analyses were performed on the otolith edges of developing and post-spawner females which were previously identified by analyzing the ovary microscopic maturity stages using histological sections (according to Brown-Peterson et al. [23]). Larger females were not considered for this analysis because otolith daily increments in large individuals are incomplete or compressed at the outer edges [24]. From the total, 35 specimens were post-spawning females (regressing-RG and regenerating-RN) and 25 specimens were pre-spawning ones (developing-D) that were used as a control group, assuming that D females capitalize all energy for gonadal growth and gamete development prior to the spawning [23].

From the selected females, six were discarded due to over-polishing of the otolith during sample processing. Accordingly, 54 females were analyzed: 23 in D, 18 in RG, and 13 in RN. Additionally, females were classified in two categories in relation to the spawning peak, based on the capture date and the ovary microscopic phase: females which present evidences of having spawn in the winter-spring spawning peak (SP1) and females with evidences of having spawn in the summer spawning peak (SP2) (Table 1).

**Table 1.** Number of sampled females (*n*) in developing (D), regressing (RG), and regenerating (RN) for the daily growth analysis considering the two first spawning peaks (SP1, SP2) and the total length range (TL range, cm).

| Ovary Phase | SP1 | | SP2 | | Total *n* |
|---|---|---|---|---|---|
| | *n* | TL Range (cm) | *n* | TL Range (cm) | |
| D | 4 | 34–73 | 19 | 49–79 | 23 |
| RG | 9 | 45–72 | 9 | 46–56 | 18 |
| RN | 6 | 47–60 | 7 | 49–58 | 13 |
| Total general | 19 | 34–73 | 35 | 46–79 | 54 |

For daily increments analysis, right sagittae otoliths were selected (Figure 2A). Then, otoliths were embedded in epoxy resin, cut in cross-sections (transversal section) and assembled with Crystalbond adhesive on a slide for subsequent polishing. Finally, using the cross-sectional preparation, sequential images of the entire ventral axis were taken at 20× magnification (Figure 2B).

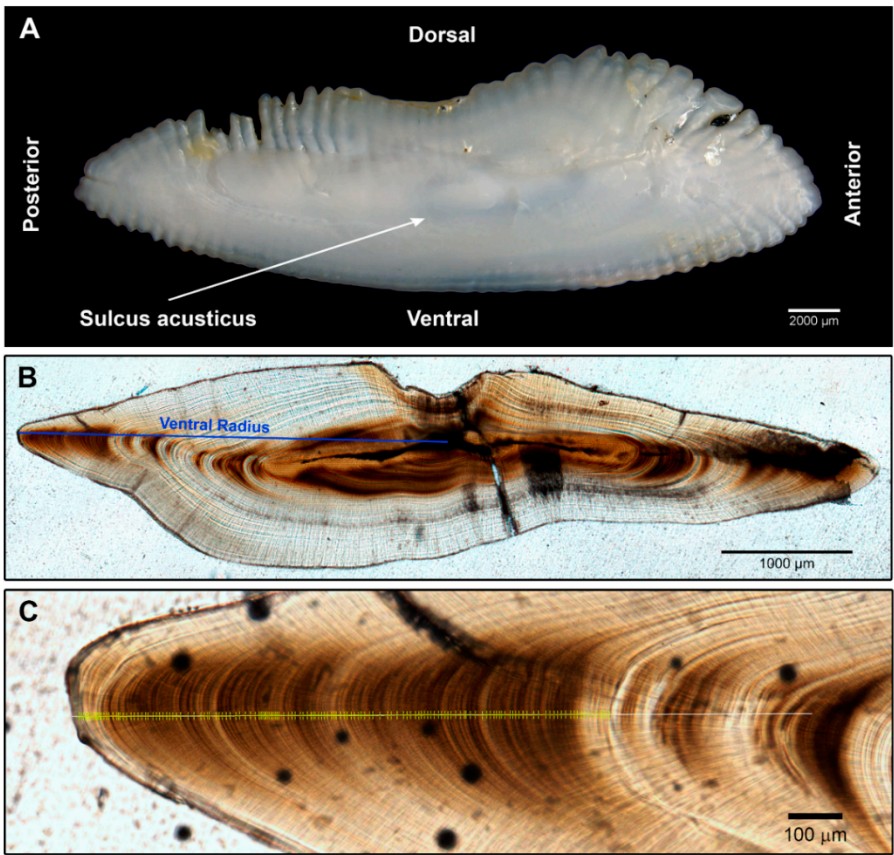

**Figure 2.** (**A**) Image of European hake right sagittae otolith (5× of magnification) before its processing. (**B**) Mosaic of the cross-section of right sagittae otolith (20× of magnification), with the ventral radius (blue line). (**C**) Axis (white line) and marks (yellow crosses) of the ventral lecture of the cross-section of the right otolith (20× magnification).

The radius was measured from the core along the ventral axis, and daily rings were counted following the same axis, from border to the nucleus of the otolith, until daily growth rings were discernible (Figure 2C).

Daily ring increments were standardized based on the otolith radius for posterior analyses to remove the female's size effect. An ANOVA was performed to compare standard distance mean between months (from January to July). As the duration of each ovarian development phase in European hake remains unknown due to its reproductive cycle, comparisons between phases were performed using only the last 15 days before the capture, assuming that females were in the same ovarian stage within the 15 days prior to capture as on the day they were caught, i.e., each ovary phase lasts at least 15 days. Additionally, an analysis of covariance (ANCOVA) was used to test for seasonal differences of the standard distance mean between otoliths of females at different ovarian development phases. Then, a post-hoc (Tukey's test) test was performed to compare between all of the levels for each factor in these analyses.

A generalized additive mixed model (GAMM) was performed to identify temporal differences in daily increments (standard distance) of the last 15 days before the capture of the female considering the capture day, ovary phases (D, RG and RN), and SP (SP1 and SP2). Capture day was included as an explanatory fixed variable but with a double

interaction: ovarian phases and SP. Moreover, female ID was included as random effect to take into account the autocorrelation inherent in the daily rings in each female. GAMM models were performed using "mgcv" R package (version 1.8.31 Simon Wood, Edinburgh, United Kingdom [25]). The model had the following structure:

$$\begin{aligned}\text{Standard distance} \\ &= \beta + f_1(\text{Capture day, interaction} = \text{SP} - \text{Ovarian phase}) \quad (1)\\ &+ \text{Random effect (ID)} + \varepsilon\end{aligned}$$

where β is the intercept and ε the residual error vector. The optimal model was selected based on the minimization of the Akaike information criterion (AIC [26]), and pairwise comparisons among factors were performed using Tukey's test.

Likewise, and because of the asynchrony of *M. merluccius*, the time series of the standard distance five-days rolling mean of each female's otoliths was examined by the naked eye in order to detect patterns in the daily growth that may be related to the reproductive cycle, and specifically to with spawning.

All statistical analyses were conducted using R software version 3.4.3 [27], with a significance level of $\alpha = 0.05$.

## 3. Results

Daily growth analysis was performed in order to identify changes in the growth pattern that could be related to active spawning with the last aim of trying to obtain individual reproductive parameters in the population. A total of 54 females with a total length range of 34–80 cm, at D, RG, and RN ovarian phases and collected both in the SP1 and SP2 were analyzed (Table S1). Otolith daily increment counts ranged from 18 to 288 by individual (with a mean of 75 read increments). Dorsal radius of the otoliths was measured in all the otoliths and ranged from 2 730 to 4 517 μm. Both inter- and intraindividual data presented high dispersion, with strong fluctuations throughout the analyzed time series.

From January to March, the mean daily increment increased 4%, from $9.91 \times 10^{-4}$ to $1.02 \times 10^{-3}$ μm. In April, growth dropped by 8% until the minimum value ($9.49 \times 10^{-4}$ μm) with similar daily mean distance in May ($9.55 \times 10^{-4}$ μm). Afterwards, the daily mean increment increased steadily almost 10% until July, when it reached the maximum value ($1.05 \times 10^{-3}$ μm) (Figure 3). Hence, monthly significant differences were detected in the daily increment ($p < 0.001$; Tukey test results in Table 2).

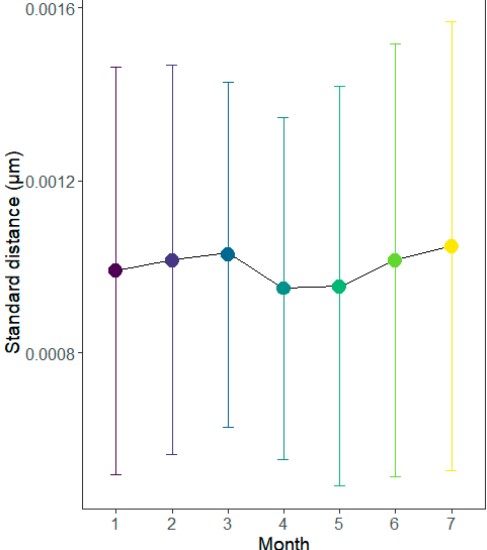

**Figure 3.** Temporal trend of the standard distance between daily rings in otoliths of European hake females captured in 2017. Points represent the mean and whiskers the standard deviation.

**Table 2.** *p*-values for Tukey multicomparison tests of the ANOVA analysis standard distance~month. Significant *p*-values are shown in boldface type.

|      | Jan. | Feb. | Mar. | Apr. | May | Jun. | Jul. |
|------|------|------|------|------|------|------|------|
| Jan. | 1.00 |      |      |      |      |      |      |
| Feb. | 1.00 | 1.00 |      |      |      |      |      |
| Mar. | 0.94 | 1.00 | 1.00 |      |      |      |      |
| Apr. | 0.86 | 0.35 | **<0.05** | 1.00 |      |      |      |
| May  | 0.92 | 0.45 | 0.05 | 1.00 | 1.00 |      |      |
| Jun. | 0.99 | 1.00 | 1.00 | 0.07 | 0.11 | 1.00 |      |
| Jul. | 0.81 | 0.99 | 1.00 | **<0.05** | 0.07 | 0.96 | 1.00 |

Additionally, mean standard distance (std dist) was analyzed by SP and ovarian development phases in females, considering only the last 15 days before capture. Females that spawned in SP1 presented lower daily increments than females that spawned in SP2. RG females in SP1 presented the lowest daily increments (std dist mean = $8.52 \times 10^{-4}$ μm), followed by D females of SP1 (std dist mean = $9.59 \times 10^{-4}$ μm) and then RN of the same SP (std dist mean = $9.85 \times 10^{-4}$ μm).

Wider rings were estimated in RN females (std dist = $1.03 \times 10^{-3}$ μm) and D females of SP2 (std dist mean = $1.09 \times 10^{-3}$ μm). Lastly, RG females from SP2 had the highest daily growth rates of all sampled females, with a standard distance mean of $1.10 \times 10^{-3}$ μm (Figure 4). Hence, the SP-ovary phase interaction showed significant influence on standard distance ($p < 0.001$; Tukey test results in Table 3).

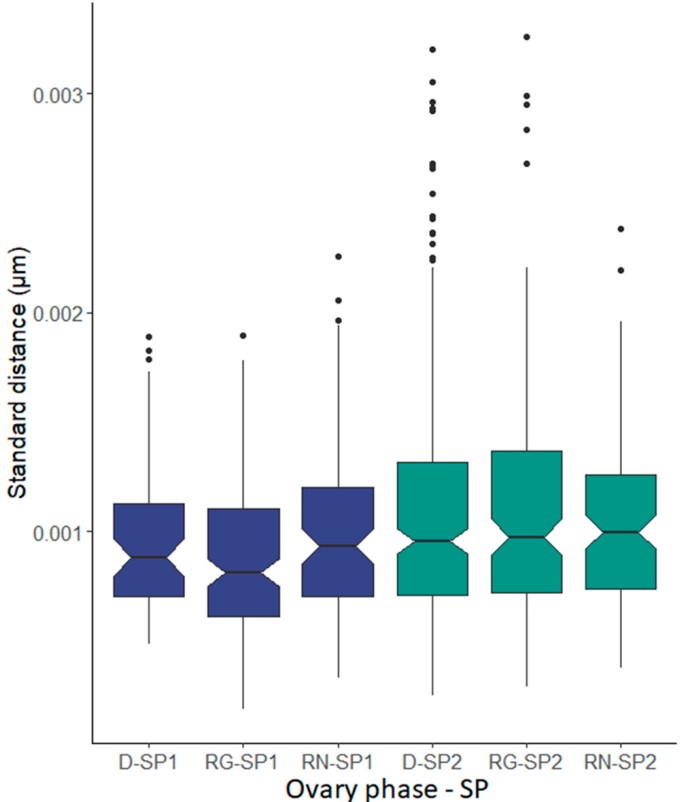

**Figure 4.** Standard distances (μm) of daily increments in the last 15 days before capture in European hake females captured in 2017 by ovarian development phase (D: developing, RG: regressing, RN: regenerating) and spawning peak (SP1, SP2). The box represents the interquartile range (25th to the 75th percentile), the line in the box represents the mean value, the solid vertical lines represents the standard error, and the notch is the 95% confidence interval of the median.

**Table 3.** *p*-values for Tukey multicomparison tests of the ANCOVA analysis standard distance~SP-ovarian development phases in the last 15 days before capture in European hake females. Significant *p*-values are shown in boldface type.

|  | **SP1-D** | **SP1-RG** | **SP1-RN** | **SP2-D** | **SP2-RG** | **SP2-RN** |
|---|---|---|---|---|---|---|
| SP1-D | 1.00 | | | | | |
| SP1-RG | 0.66 | 1.00 | | | | |
| SP1-RN | 1.00 | 0.27 | 1.00 | | | |
| SP2-D | 0.32 | **<0.001** | 0.39 | 1.00 | | |
| SP2-RG | 0.31 | **<0.001** | 0.38 | 1.00 | 1.00 | |
| SP2-RN | 0.92 | **<0.05** | 0.98 | 0.87 | 0.83 | 1.00 |

The GAMM performed with the standardized daily increments of last 15 days of life in sampled females showed different effects of ovarian development phases and SP on the standard distance. Not all SP-ovary phase combinations had a significant effect on the daily increment: in RG and RN, females from both SP standard distances significantly increased as days progress ($p_{SP1-RG} < 0.05$, $p_{SP1-RN} < 0.05$, $p_{SP2-RG} < 0.001$, $p_{SP2-RN} < 0.05$) while in D, females from both SP1 and SP2 did not show significant variation of standard distance with time ($p_{SP1-D} = 0.24$, $p_{SP2-D} = 0.15$) (Table 4). Figure 5 shows the additive effect of the capture day on the standard distance: D females did not present any pattern in either of the two spawning peaks. Conversely, post-spawning females (RG and RN) of SP1 and SP2 showed an upward trend, which appears more intense in the SP2-RG group.

**Table 4.** Parameter estimates by the fixed effects of the generalized additive mixed model (GAMM) of daily increments in the last 15 days before capture in developing (D), regressing (RG) and regenerating (RN) European hake females captured in 2017. Parametric coefficients present: estimate, standard error (Std. error), z value, and *p*-value; smooth terms present: estimated degrees of freedom (edf), standard error (Std. error), z value, and *p*-value. Moreover, total Akaike Information Criterion (AIC) was calculated.

| **Parametric Coefficients** | **Estimate** | **Std. Error** | *t* **Value** | *p*-**Value** |
|---|---|---|---|---|
| Intercept | −6.94 | 0.05 | −152.50 | <0.001 |
| **Smooth Terms** | **edf** | **Ref.df** | **Chi.sq** | *p*-**Value** |
| SP1–D | 1.00 | 1.00 | 1.37 | 0.24 |
| SP1–RG | 1.00 | 1.00 | 4.95 | <0.05 |
| SP1–RN | 1.00 | 1.00 | 4.30 | <0.05 |
| SP2–D | 1.00 | 1.00 | 2.12 | 0.15 |
| SP2–RG | 1.99 | 1.99 | 7.44 | <0.001 |
| SP2–RN | 1.00 | 1.00 | 3.93 | <0.05 |
| | | AIC = 531.40 | | |

GAMM was associated with population asynchrony, thus, individual trends were analyzed by the naked eye to identify possible patterns associated with spawning: a relationship between spawning and the daily increment. In some females was detected a period of one to two months during which 4–6 valleys of narrow daily increments appear. The described pattern was not repeated over time—it appears once in each female sampled—and did not overlap on the same dates in females due to population asynchrony. Only 15 females—28% of the sampled females—presented this pattern of daily increment variability that may be related with spawning, more specifically 7 females from SP1 (5 RG and 2 RN) and 8 females from SP2 (3 RG, 4RN and 1 D) (Figure S1 represents the time series of the sampled females with the detected pattern). In general, the expected pattern of daily increments that could be linked to spawning activity was not observed in the otoliths of the largest females (>60 cm), excepting one female of 72 cm, despite sampled females reaching sizes of 80 cm.

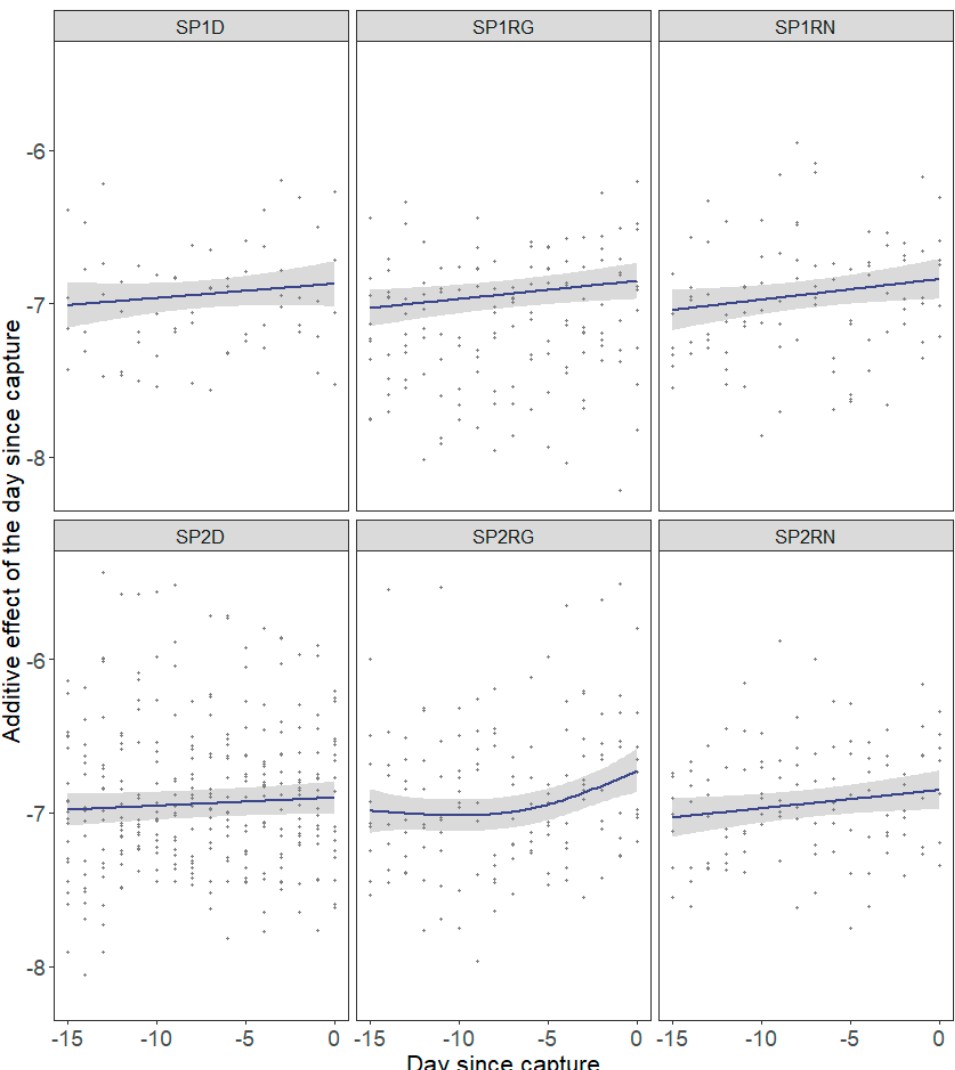

**Figure 5.** Response plots on the partial effects on the generalized additive mixed model (GAMM) of daily increments in the last 15 days before capture in European hake females captured in 2017. Plot includes the additive effect spawning peaks (SP1, SP2) and ovarian development phase (D, RG, RN). Solid lines denote smoothed values and shaded areas represent 95% confidence intervals.

## 4. Discussion

Spawning dynamics, both in terms of space and time, is a critical component in reproductive success because of its effect on development and the survival of the offspring and hence on recruitment [28–30]. Specially, spawning time has long been revealed as a one of the most relevant factors [31]. Compared with other areas, European hake in the Iberian shelf has a very protracted spawning season, likely an adaptation to the highly productive but also highly variable environmental conditions in this area [10,22]. Indeed, the intra-annual variability in spawning timing has been also evidenced in other species of *Merluccius*, and the existence of numerous peaks of spawning was associated to different stocks or sub-stocks [32–34]. Hake in the study area presents several spawning peaks, but it is difficult to elucidate its reproductive dynamics because the analysis of maternal attributes does not provide clear evidences. Likewise, the impossibility of captive breeding hinders the estimation of individual reproductive parameters. Therefore, given that otoliths are records of fish life and daily growth is expected to be linked to reproductive activity, we tested whether patterns in daily growth can be identified that can be related to individual spawning (onset, duration, frequency, or subcomponents).

Considering the trade-off between growth and reproduction, the oocyte dynamics may impact growth performance in European hake [35]. The initial assumption considers a lower growth in females when they are spawning, from the oocyte recruitment until the egg release, because gamete production and reproductive behaviour have a cost in terms of energy [2]. Several studies have detected the presence of discontinuities in the width pattern of annual otolith increments which are associated to reproductive processes, such as the onset of sexual maturity (only once in the lifetime of the individual [36–38]) or spawning activity [39]. This last study was focused on the relationship between translucent and opaque zones in the otoliths, but results were not conclusive. However, as active reproduction is expected to lead to low growth rate, the hypothesis of the present study is that daily increments in otoliths will be narrower during the spawning peak than outside it. The analysis is based on two assumptions. First, pre-spawner females capitalize most of their energy intake in oocyte development for spawning, and second, females that have spawn (RG and RN) devote most of their surplus energy to growth. The result is that in RN, females the daily increments at the edge of the otolith should reflect an accelerated growth pattern after a period of low daily growth, during spawning.

Temporal differences in daily growth were detected in our analyses, with narrow rings in April–May and wide ones in July–August, already detected in juveniles, i.e., immature fish, of the same sampling area [40]. This result associated to the temporal dynamics showed that daily growth may be more influenced by other factors than the impact of reproduction, such as environmental conditions. Seasonal differences in the Galician Shelf with cooler waters on winter—spring than the rest of the year in association to the other oceanographic conditions of the area [41,42] probably influenced the growth result of slower daily growth rates [43].

Regarding mature females, the individuals with the lowest daily growth increments were those in regressing phase after the winter—spring spawning peak. In addition to the environmental conditions, narrower daily rings may be associated with a spawning event. In concordance with previous studies, spawners from winter—spring present higher reproductive effort than spawners from the rest of the years, i.e., fecundity, egg diameter, or dry weight are higher in that spawning peak than in the summer and autumn [10,22,44,45]. So, the acquired energy may be devoted to ensuring maintenance and recovery of body mass, rather than length, after an exhausted spawning period.

However, since then, daily growth steadily increased in all the post-spawning females and the highest daily growth rates are recorded in the regressing females from the summer spawning peak. In contrast to the spring—winter peak, the summer spawning peak with less reproductive effort allows for a greater growth of the females. As well, upwelling events during summer enhances food availability due to higher primary production, and consequently, higher energy acquisition for growing [46,47].

This indicates a complex and intricate trade-off between the feeding, growth, and reproductive dynamics. However, some limitations in this analysis may hinder the temporal variation between ovarian development phases in the daily increments, such as the high dispersion of the data and the inability to determine the duration of each ovary phase. Nevertheless, our modelling approach that includes temporal variability may shed light into the matter of growth dynamics related with spawning. Developing females caught both in the winter—spring and summer spawning peak did not show any trend in daily growth, likely because these females maintain basal growth and invest the entire surplus in reproduction [23]. On the other hand, post-spawning females in both peaks had an accelerated growth, evidencing the recovery after the energy expenditure in spawning. Nevertheless, the daily increments showed large inter- and intraindividual variability, making it difficult to reach robust conclusions. As previously mentioned during the study, the large variability, the uncertainty about the duration of each ovary phase throughout the year, and the high asynchrony of the population hamper the analysis of the hake spawning dynamics in determining the individual reproductive behaviour exactly, but it provides information that allows us to observe certain individual spawning patterns.

One particular and interesting pattern observed in the daily ring increments time series is the existence of regular intervals of decreasing growth rate every four to five days in several females which as a "spawning pattern" could be related to the spawning events (batch release) and thus be a reflection of the spawning frequency. In fact, this is the first report of an approximation of individual spawning frequency in European hake.

This "spawning pattern" in the otolith was discernible only in 15 females (28%), all of them with medium and large size. Small females likely invest less in reproductive effort and still require a major investment in growth. Rollefsen (in 1933) [48] also found spawning-related growth patterns in 60% of the analyzed otoliths of cod in Lofoten based on annual rings and suggested that younger females showed minor changes in daily growth during spawning because the energy invested in reproduction was lower. In other words, if the energy investment does not compromise female's growth, a spawning pattern cannot be detected in the otolith because of the absence of spawning checks. A similar pattern, but on a daily basis, can be expected in young female hake, explaining the absence of the "spawning pattern" in the otoliths.

Daily growth dynamics analysis in European hake showed a relationship with the spawning dynamics, but is not sufficiently conclusive about the number of spawning components. This study has shed light on the study of the population dynamics of this stock: the pattern is detected only once per year in the otolith, so a female would participate in only one spawning peak. The presence or two or more spawning components seems to better explain the observed pattern in growth and spawning dynamics of European hake and the involved trade-offs, as was hypothesized in previous works of this stock [22,44]. Thus, the next step may be the analysis of the contribution of each spawning subcomponent to the annual recruitment in the Southern stock of hake.

Several spawning-related factors may impact daily growth, such as spawning vertical migrations, seasonal habitat changes for spawning (different environmental conditions), or the reduction of food intake during the egg release. Other factors, as fluctuations in prey availability or reduction in metabolism (e.g., parasites or diseases) will certainly cause alteration in growth rate [48–50], but not in a rhythmic pattern as observed.

A part of the knowledge gap existing on the otolith formation relates to energy trade-off [50]; one constraint on using otolith daily growth to correlate with spawning activity is the fact that the detection of daily rings in adult females is laborious and complex. It has been shown that the number of daily rings decreases and becomes less discernible with age [24,51,52]. Moreover, the detection was made with the underlying assumption of a constancy in the daily increment periodicity. However, the existence of sub-daily and even the absence of daily rings make the otolith increments interpretation difficult [24,53]. Therefore, a validation of daily increments and aging criteria in adult hake is necessary for a better interpretation of data.

## 5. Conclusions

In spite of the large variability recorded in the daily growth patterns of European hake in the Galician waters, an evident daily growth pattern coincident with the expected spawning pattern is observed in some females, indicating that individual spawning season lasts 1.5–2 months and that spawning frequency is 4–5 days. In fact, this is the first report of an approximation of individual spawning frequency in European hake. Thus, this study highlights the usefulness and potential of daily growth analysis in the estimation of individual reproductive parameters. In combination with otolith microchemistry or isotopes, the knowledge of the reproductive parameters of exploited species may improve notably, especially when the species presents population asynchrony.

**Supplementary Materials:** The following supporting information can be downloaded at: https: //www.mdpi.com/article/10.3390/fishes7040208/s1, Table S1: Data from the daily growth analysis of the adult females of European hake captured in 2017. A total of 54 females were analysed: 23 developing (D) females, 18 regressing (RG) females and 13 regenerating (RN) females. Each row corresponds to one sampled female, SP to the assumed spawning peak in which each female spawn (SP1 or SP2), capture date, TL to the total length of each female (in cm), otolith radius to the ventral radius (in μm), *n* rings the number of rings that were able to be read and Std distance, the standard distance mean between daily increments (and the standard deviation) (in μm). Figure S1: Rolling mean standard distance (μm) of daily increment time series (of the last 150 days of life) in European hake females captured in 2017 with a detected spawning pattern: period with 4–6 valleys of reduced daily growth extending over 30–60 days (period marked in the figure between vertical dashed grey lines). Dark blue represents females which spawn in SP1 (Dec–Mar) and dark green represents females which spawn in SP2 (Apr–Jul). Ovary phase (D: developing, RG: regressing, RN: regenerating) and total length (cm) are showed in the upper part of each plot. Note the difference in the y axis.

**Author Contributions:** Conceptualization, formal analysis and methodology, C.G.-F.; validation, R.D.-P.; investigation, C.G.-F., R.D.-P. and F.S.-R.; resources, F.S.-R.; writing—original draft preparation, C.G.-F.; writing—review and editing, R.D.-P. and F.S.-R.; supervision, R.D.-P. and F.S.-R.; project administration, F.S.-R.; funding acquisition, F.S.-R. All authors have read and agreed to the published version of the manuscript.

**Funding:** C.G.-F. was funded by Predoctoral Fellowship from the Fundación Tatiana Pérez de Guzmán el Bueno and the study was carried out with financial support from the DREAMER project (CTM2015-66676-C2-1-R) funded by the Spanish National Research Program.

**Institutional Review Board Statement:** Ethical review and approval were waived for this study, because the described scientific sampling did not require ethical permission according to the applicable international, EU and national laws.

**Informed Consent Statement:** Not applicable.

**Data Availability Statement:** The data that support the findings of this study are available from the corresponding author upon reasonable request.

**Acknowledgments:** Special thanks to Lorena Rodríguez, Carmen Piñeiro, Maria Sainza, Santiago Cerviño and Antonio Gomez from Centro Oceanográfico de Vigo (IEO-CSIC) for providing samples and for suggestions for improving the study.

**Conflicts of Interest:** The authors declare no conflict of interest.

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
