# Peer review of "The Use of Daily Growth to Analyze Individual Spawning Dynamics in an Asynchronous Population: The Case of the European Hake from the Southern Stock†"

_fishes, doi:10.3390/fishes7040208_

Round 1

Reviewer 1 Report

Great work, well-written paper, and very useful study. Suggestion. I suggest that the text from line 66-71 and line 73-91 mix and concisely write the hypotheses and objectives of the paper and put them in the introduction.

Author Response

We thank the effort and time invested by the reviewer as well the valuable comments provided. Moreover, the end of the introduction was changed in the manuscript for a better understanding, mixing both text parts from line 68 to line 75.

Reviewer 2 Report

Spawning dynamics analysis of the European hake (daily growth of otoliths) is very important in order to understand and manage the reproductive phase.

The pattern observed in the daily ring increments time series is very interesting and revelant to the spawing events. Daily growth pattern is coincident with the expected spawning pattern.

Manuscript is clear and well organized.

Statistical analysis is sound.

Author Response

We thank the referee the positive assessment of our study as well as his/her interest on the importance of this analysis.

Reviewer 3 Report

The paper describes the daily growth pattern and its relationship with reproduction was analyzed in European hake from the Galician Shelf. Overall, I appreciated the work both for the results achieved and, in general, for the organization. However, I believe that changes in the methodological part are needed before the paper can be accepted for publication. My comments are outlined below.

line 28: if possible, please add a reference

line 73-91: I am not convinced that this part fits in this paragraph. I would suggest that the authors move it to the INTRODUCTION paragraph and integrate it with lines 66-71

line 99: there is no reference to the area of origin of the samples. it would be useful to include this information and perhaps provide a map, especially for readers who are unfamiliar with the area. there is also no indication on how the specimens were obtained for analysis (scientific fishing? professional fishing? other?)

line 218: there is a problem with the reference

Author Response

Firstly, we appreciate the time and effort delivered by the referee to review this manuscript as well as his/her comments. Suggestions and comments have improved it significantly. Below you can find our responses (in green)

The paper describes the daily growth pattern and its relationship with reproduction was analyzed in European hake from the Galician Shelf. Overall, I appreciated the work both for the results achieved and, in general, for the organization. However, I believe that changes in the methodological part are needed before the paper can be accepted for publication. My comments are outlined below.

line 28: if possible, please add a reference

Added. We added the following reference: M. Heino and V. Kaitala, “Evolution of resource allocation between growth and reproduction in animals with indeterminate growth,” J. Evol. Biol., pp. 423–429, 1999

line 73-91: I am not convinced that this part fits in this paragraph. I would suggest that the authors move it to the INTRODUCTION paragraph and integrate it with lines 66-71

Corrected. Change made based on your comment and another reviewer's comment. The text has improved and is much clearer now.

line 99: there is no reference to the area of origin of the samples. it would be useful to include this information and perhaps provide a map, especially for readers who are unfamiliar with the area. there is also no indication on how the specimens were obtained for analysis (scientific fishing? professional fishing? other?)

Corrected. We totally agree that there was a lack of information in the source of samples and we added a short explanation from line 99 to line 105. However, we think that is better a description than a map due to the amount of figures in the manuscript. In addition, a short explanation of the study area is already presented from line 54 to line 59.

line 218: there is a problem with the reference

Corrected. We remove the reference; it was a mistake (a hyperlink with a Figure)

Reviewer 4 Report

The manuscript contains an interesting research on the context of growth and spawning of European hake based on otolith microstructure. The subject of the paper is worthy of investigation.

The introduction is appropriate.

Methods and experimental design seemed to be correct, but there are some issues to be clarified on the methods and the presentation of the results

Material and methods:

Sample collection: It is not clear how the seven months were covered in the study. The capture date starts in February and only some samples were collected in March and April (based on the supplementary material)

Statistical analysis: If I understand the study correctly, the ANCOVA and the GAMM examine the same. Why do you use two different tests? Additionally, please add more details on ANCOVA (what was the model- variables and covariates?).

Results:

Please add the details of the ANCOVA. It would be nice to see which factors and also their interactions were significant. I think that in this case, less could be more. The sample size is not too high but the number of groups is. If the data were evaluated based on fewer criteria, I think a clearer picture could be obtained (maybe SP1 and SP2 should be merged).

To me, what you can see is that RG (and RN) females and D females are different. Since the research aimed to prove this, it should be clearly supported by statistics.

Minor suggestions:

Lines 80-91 should be moved to the introduction.

Table 2 March-May is also significant.

Line 280 August and October -November were not sampled in this study! – please rephrase it.

Author Response

First of all, we want to express our gratitude to you for the time devoted to our manuscript. We appreciate your new suggestions to improve the manuscript. Below you can find our responses to your comments (in green)

The manuscript contains an interesting research on the context of growth and spawning of European hake based on otolith microstructure. The subject of the paper is worthy of investigation.

The introduction is appropriate.

Methods and experimental design seemed to be correct, but there are some issues to be clarified on the methods and the presentation of the results

Material and methods:

Sample collection: It is not clear how the seven months were covered in the study. The capture date starts in February and only some samples were collected in March and April (based on the supplementary material)

Added. We include the sampling coverage and more description about samplings from line 99 to line 102. Sampling was performed along the whole year (2017) each month with higher frequency in the first part of the year.

Statistical analysis: If I understand the study correctly, the ANCOVA and the GAMM examine the same. Why do you use two different tests? Additionally, please add more details on ANCOVA (what was the model- variables and covariates?).

The analysis was not the same: with the ANCOVA, the analysis was without considering specific days while in the GAMM; day since capture was included as a variable in the model, to see the growth curve of the individual in the 15 days before the capture as well as the random effect of each sampled female.

Moreover, the details of the ANCOVA are already explained in the text from line 152 to line 154, showing the model variables (standard distance mean ~ spawning peak * ovarian development phase).

Results:

Please add the details of the ANCOVA. It would be nice to see which factors and also their interactions were significant. I think that in this case, less could be more. The sample size is not too high but the number of groups is. If the data were evaluated based on fewer criteria, I think a clearer picture could be obtained (maybe SP1 and SP2 should be merged).

To me, what you can see is that RG (and RN) females and D females are different. Since the research aimed to prove this, it should be clearly supported by statistics.

We think that it is better to show Tukey test results from that ANCOVA because, as it is explained in lines 206 and 207, the SP-ovary phase interaction showed significant influence on standard distance (p < 0.001)

Minor suggestions:

Lines 80-91 should be moved to the introduction. Moved to Introduction (from line 75 to line 80)

Table 2 March-May is also significant. This result is not significant as it is greater than 0.05, but is rounded to two decimal numbers

Line 280 August and October -November were not sampled in this study! – please rephrase it. Removed from the text.

Round 2

Reviewer 4 Report

I accept corrections and responses to my suggestions.